# Antileishmanial Compounds Isolated from *Psidium Guajava* L. Using a Metabolomic Approach

**DOI:** 10.3390/molecules24244536

**Published:** 2019-12-11

**Authors:** Chiobouaphong Phakeovilay, Sandra Bourgeade-Delmas, Pierre Perio, Alexis Valentin, François Chassagne, Eric Deharo, Karine Reybier, Guillaume Marti

**Affiliations:** 1UMR 152 Pharma Dev, Université de Toulouse, IRD, UPS, 31000 Toulouse, Francesandra.bourgeade-delmas@ird.fr (S.B.-D.); pierre.perio@univ-tlse3.fr (P.P.); alexis.valentin@univ-tlse3.fr (A.V.); francois.chassagne@emory.edu (F.C.); eric.deharo@ird.fr (E.D.); karine.reybier-vuattoux@univ-tlse3.fr (K.R.); 2Faculty of Pharmacy, University of Health Sciences, Ministry of Health, Vientiane 01000, Lao People’s Democratic Republic; 3Center for the Study of Human Health, Emory University, 615 Michael Street, Whitehead Building, Atlanta, GA 30322, USA

**Keywords:** metabolomic, antileishmanial activity, metabolomics, *Psidium guajava*, jacoumaric acid, corosolic acid

## Abstract

With an estimated annual incidence of one million cases, leishmaniasis is one of the top five vector-borne diseases. Currently available medical treatments involve side effects, including toxicity, non-specific targeting, and resistance development. Thus, new antileishmanial chemical entities are of the utmost interest to fight against this disease. The aim of this study was to obtain potential antileishmanial natural products from *Psidium guajava* leaves using a metabolomic workflow. Several crude extracts from *P. guajava* leaves harvested from different locations in the Lao People’s Democratic Republic (Lao PDR) were profiled by liquid chromatography coupled to high-resolution mass spectrometry, and subsequently evaluated for their antileishmanial activities. The putative active compounds were highlighted by multivariate correlation analysis between the antileishmanial response and chromatographic profiles of *P. guajava* mixtures. The results showed that the pooled apolar fractions from *P. guajava* were the most active (IC_50_ = 1.96 ± 0.47 µg/mL). Multivariate data analysis of the apolar fractions highlighted a family of triterpenoid compounds, including jacoumaric acid (IC_50_ = 1.318 ± 0.59 µg/mL) and corosolic acid (IC_50_ = 1.01 ± 0.06 µg/mL). Our approach allowed the identification of antileishmanial compounds from the crude extracts in only a small number of steps and can be easily adapted for use in the discovery workflows of several other natural products.

## 1. Introduction

Leishmaniasis is in the top five of vector-bone diseases identified in the Global Health Estimates 2016 summary table of the World Health Organization (WHO). It is one of 20 neglected tropical diseases that infect one billion people in low socioeconomic populations in 149 countries [1]. Leishmaniasis is caused by the genus *Leishmania* spp., which includes 29 species. Among them, *Leishmania donovani* and *Leishmania infantum* cause visceral leishmaniasis, the lethal form of the disease [2]. Leishmaniasis is transmitted to mammalians by the bites of the female phlebotomine sandfly. The epidemiology of leishmaniasis depends on a variety of factors including the between host, reservoir and vector (human, animal and sandfly), the local ecological characteristics of the transmission sites such as alterations in temperature and water storage, irrigation habits, deforestation, climate changes, exposure of the human population to the parasite, and human behavior [3]. The disease is characterized by four forms; visceral leishmaniasis, post-kala-azar dermal leishmaniasis, cutaneous leishmaniasis, and mucocutaneous leishmaniasis [4]. Available medical treatments are not sufficient because of their extensive toxicity (e.g., the nephrotoxicity of amphotericin B, the teratogenicity of miltefosine), their lack of efficacy, their expensive cost (amphotericin B), the development of drug resistance (sodium stibogluconate and meglumine antimoniate), and non-specific targeting [4,5]. Fexinidazole, a 5-nitroimidazole, was recently in phase II of clinical trials on visceral leishmaniasis but exhibited a lack of efficacy, and in 2018 no new lead molecules were included in clinical trials [4].

Control of leishmaniasis by traditional approaches (vector control and treatment of patients) is facing the difficulties of identifying new cost-effectively produced bioactive molecules able to be produced on an industrial scale, rapid development of drug resistance, and the lack of access to drugs by the threatened populations. These issues are at the heart of concerns of the WHO regarding the control of tropical diseases [6]. Therefore, combination therapies are used to improve the efficacy of antileishmanial therapies and can shorten treatment duration [7]. Furthering our fundamental knowledge through original approaches, herein concerning the activity of natural compounds from a promising plant species active against *Leishmania* spp., is vital to overcome the challenges involved in treating leishmaniasis [8].

*Psidium guajava* Linn., or guava, is a tree belonging to the Myrtaceae family. Guava is native to Mexico [9], and mainly grows in tropical or subtropical areas. Guava is widely used for food and traditional medicine around the world. The guava leaves can be used in the form of an infusion or decoction for the treatment of diarrhea (especially in Lao People’s Democratic Republic (PDR) [10]), pain, fever, cough, diabetes, hypertension, wounds [9], spasms, and rheumatism [11]. Biological assays revealed that leaf extracts exhibit antibacterial and antifungal activity against *Staphylococcus aureus*, *Streptococcus pyogenes*, *Escherichia coli*, *Pseudomonas aeruginosa*, and *Candida albicans* [12]. Previous studies have described its antileishmanial potential, showing a 90% and 75% inhibition of *L. donovani* and *L. infantum* promastigotes, respectively [13,14]. However, to date, antileishmanial activity of purified active compounds from guava has not been reported.

The purpose of this study was to find new antileishmanial compounds from leaf extracts of *P. guajava* L. using a metabolomic approach and correlation analysis. Our group has previously described the use of this workflow to decipher the most active redox compounds in crude extracts of *Viola alba* subsp. *Dehnhardii*, Violaceae [15] and to identify antihepatocarcinogenic compounds from plants used to treat liver cancer in Cambodia [16]. Our approach aims to link the chemical profile variability of several extracts or fractions to bioassay results using multivariate data analysis. This holistic method has been implemented to rapidly understand the diversity and role of chemical components involved in bioactivity and is in contrast to the reductionist approach that involves successive fractionation steps for the purification of compounds responsible for the activity [17]. The use of partial least square (PLS) regression models allowed the ranking of putative active features detected from liquid chromatography–mass spectrometry (LC–MS) profiles. The annotation of peaks of interests was based on high-resolution mass spectrometry (HRMS) and tandem mass spectrometry (MS/MS) patterns mirrored to in silico fragmentation and confirmed by purification or commercial authentic standards [18].

## 2. Results

### 2.1. Antileishmanial Activity

Antileishmanial activities of the nine crude extracts and fractions from *P. guajava* leaves collected in different areas of Lao PDR were tested on the promastigote and amastigote forms of *L. infantum* at concentrations of 25 and 50 µg/mL. The results indicated that crude extracts, and polar and apolar fractions were not active on promastigote forms, whereas crude extracts and apolar fractions inhibited over 50% of *L. infantum* amastigotes at 25 µg/mL (Figure 1A). The apolar fractions showed the highest antileishmanial potential with more than 80% growth inhibition. The IC_50_ values of the nine apolar fractions ranged from 1.62 ± 0.04 µg/mL to 2.886 ± 0.24 µg/mL (Figure 1B). These results are not so far from the values obtained with amphothericin B, taking into account that we assessed an extract and not a pure molecule (Figure 1B).

### 2.2. UHPLC–HRMS-based Metabolomics Approach

UHPLC–HRMS (ultra-high-performance liquid chromatography–high resolution mass spectrometry) profiles of nine extracts (three crude extracts from the Champasak province, three crude extracts from the Savannakhet province, and three crude extracts from the Vientiane province (Figure 2A) provided 448 features (*m*/*z*-RT pairs) in negative ionization (NI) mode and 163 features (*m*/*z*-RT pairs) in positive ionization (PI) mode. To obtain an unsupervised overview of sample closeness, a principal component analysis (PCA) score plot was performed to project crude extracts and quality control (QC) samples (an aliquot of each sample used in the study) on the latent variable space. This PCA score plot revealed that the analytical workflow was accurate and reproducible due to the central clustering of QC samples. In addition, samples did not cluster according to collection areas; only two samples from the Champasak province did not cluster with the other samples (Figure 2B(a)).

We then built an orthogonal projection to latent structures (OPLS) regression model to rank the features (*m*/*z*-RT pairs) toward the IC_50_ values of antileishmanial activity (Y-input) for polar and apolar fractions. The quality of model prediction was adequate (R2Y = 0.985, Q2Y = 0.97) and a permutation test assessed its validity (Appendix A). The model revealed that the liquid–liquid extraction method clearly separated polar and apolar fractions in terms of chemical constituents and antileishmanial activity, and the apolar phases showed the most promising antileishmanial potential (Figure 2B(c)). This supervised technique classified potentially antileishmanial compounds according to their coefficient values (Figure 2C(a,b)), with positive coefficients relating to a high correlation with antileishmanial potential and negative coefficients to non-active compounds. The first seven hits (Figure 2C(c)) were identified based on their respective accurate mass and MS/MS analysis (Appendix A).

### 2.3. Identification of Putative Antileishmanial Compounds Based on Liquid–liquid Extraction

The first seven ranked compounds from the OPLS regression analysis were putatively annotated by matching compounds in the Dictionary of National Products (DNP) database (CRC press, v27.2). Compounds belonging to the Myrtaceae family and the *Psidium* genus were prioritized to narrow down the possibilities. For each compound, some candidates were ranked and proposed based on their similarity score according to the comparison between experimental MS/MS fragmentation and the in silico spectra of candidates. This process resulted in the annotation of seven candidates belonging to the *Psidium* genus, and all were classified as triterpenoid compounds. Annotation results were supported UV spectra of each peak (Table 1; column “UV”). The UV spectra of three triterpenoids (compounds **2**, **4**, **5**) showed an absorption band between 278 and 310 nm highlighting the presence of a conjugated radical (Figure 3). If available, commercial authentic standards were used to confirm feature annotation (compounds **2**, **5**, **6**) or purified from crude apolar fraction and confirmed by 1D and 2D nuclear magnetic resonance (NMR) (compound **1**, Appendix A). 

### 2.4. Putative Mechanism of Action

Antileishmanial assays revealed that pooled apolar fractions had a good activity with an IC_50_ value of 1.96 ± 0.47 µg/mL and selectivity index (SI) of around 26. The first two ranked compounds, corosolic acid (**1**) and jacoumaric acid (**2**) were more active than guajadial B (**5**) and medicagenic acid (**6**), with an IC_50_ of 1.32 ± 0.59 µg/mL for jacoumaric acid and 1.01 ± 0.06 µg/mL for corosolic acid (Table 2).

Since most antileishmanial drugs act by the production of reactive oxygen species (ROS) that lead to the death of parasites [23,24], we measured the production of ROS induced by jacoumaric acid and corosolic acid at the IC_50_ (1.32 µg/mL and 1.06 µg/mL, respectively) and IC_90_ (1.90 µg/mL and 1.43 µg/mL, respectively) using H_2_DCFDA (2′,7′-dichlorodihydrofluorescein diacetate). The corresponding graph presented in Figure 4A clearly demonstrates that jacoumaric acid and corosolic acid have an antioxidant capacity, since the fluorescence proportional to the production of ROS significantly decreased compared to the control by addition of either compound regardless of the concentration used. We also measured the IC_50_ of these compounds when associated with mannitol, a strong antioxidant (Figure 4B). We found no significant differences in IC_50_ values, confirming that the activity of jacoumaric acid and corosolic acid does not involve the production of ROS. 

## 3. Discussion

The aim of this study was to identify antileishmanial compounds from the leaves of *P. guajava* and their putative mechanism of action. According to the inhibition percentage at 25 µg/mL, the ethanolic leaf extracts of *P. guajava* killed more than 50% of parasites, which is more than the reported antileishmanial activity of seven Brazilian plant species [25] and plants from the northwest of Morocco [26]. Moreover, the apolar fractions have interesting antileishmanial activity, with IC_50_ values ranging from 1.62 ± 0.46 µg/mL to 2.89 ± 0.49 µg/mL and an SI measured on *L. infantum* amastigotes of about 26. This result indicates that this plant is more active than *Mangifera indica* and *Digera muricate*, plants that have previously been reported to have antileishmanial activity [27]. To identify compounds responsible for this activity, we compared the in vitro antileishmanial activity and chromatographic profiles of several *P. guajava* leaf extracts using a metabolomic workflow. The approach depicted here allowed the ranking of detected features based on their correlation to a putative antileishmanial activity in a small number of steps. The annotation results were confirmed by commercial authentic standards (Table 1, compounds **2**, **5**, and **6**) or a purification process (**1**). Guajadial B exhibited antitumor activity in five human cancer cell lines (HCT116, CCRFCEM, DU145, Huh7, and A549) at an IC_50_ value of 150 nM [28]. Medicagenic acid has been identified as an antifungal agent against *Pyricularia pryzae* (minimum inhibitory concentration (MIC) = 0.01 mg/mL and minimum fungicidal concentration (MFC) = 0.03 mg/mL) [29]. In our study, corosolic acid and jacoumaric acid displayed the most interesting antileishmanial activity in the amastigotes stage with respective IC_50_ values of 1.01 ± 0.06 µg/mL and 1.32 ± 0.59 µg/mL and an SI above five. This makes these two compounds more active than other triterpenoids such as ursolic acid (IC_50_ value 27 µg/mL) [30], E-caryophyllene (IC_50_ value 10.7 ± 0.6 µg/mL) [31], and spergulin-A (IC_50_ value 6.22 µg/mL) [32]. Both compounds have previously been identified in *P. guajava* and suggested to be good candidates for the development of new treatments for sickle cell anemia [33]. Furthermore, we have demonstrated their role in ROS scavenging activity, with antioxidant capacity comparable to rosmarinic acid and caffeic acid [34]. Corosolic acid has been shown to prevent inflammation and hypertension in rats [35] and to reduce the plasma glucose level in humans [19]. Corosolic acid is also known for its antibacterial activity against *Paenibacillus larvae* or *Melissococcus plutonius* [36], its anticancer activity [37], and its antitumor effect on human cervical adenocarcinoma Hela cells [38]. Jacoumaric acid also displayed antitumor effects in many carcinoma cell lines, such as leukemia [39], colorectal cancer, and human breast cancer cell lines [40]. To the best of our knowledge, this is the first report demonstrating antileishmanial activity of both compounds at the amastigotes stage. However, the close derivatives maslinic and oleanoic acid displayed antileishmanial activity on the promastigote and amastigote forms of *L. infantum* and *L. amazonensis* in the same concentration range as recorded in this study (IC_50_ on *L. infantum* amastigote: 0.99 ± 0.09 µg/mL and 2.91 ± 0.066 µg/mL, respectively) [41]. This could highlight a common mechanism of action of these triterpenoid compounds, probably related to ROS scavenging and radical positioning. According to our biological assay results and those obtained by the teams of Torres-Santos et al. and Sifaoui et al. [30,41], the presence of a carboxy group in position C-4 decreases the antileishmanial activity. Overall, the metabolomic workflow depicted in this study allowed us to rapidly target compounds responsible for the antileishmanial activity and limit multiple fractionation steps.

## 4. Materials and Methods

### 4.1. Plant Material

On November 2016, nine samples of *P. guajava* L. (Pg) leaves were collected from middle and southern parts of Lao PDR (Champasak: Pg1 to Pg3, Savannakhet: Pg4 to Pg6 and Vientiane province: Pg7 to Pg9). Samples were washed and air-dried, protected from the sunlight, before being ground into powder to obtain 1 kg of each. A sample specimen of each sample was collected and deposited at the herbarium of the Institute of Traditional Medicine of Lao, Vientiane, Lao PDR.

### 4.2. Leaf Extraction

Each accession of *P. guajava* leaves (250 g) sample were extracted two times using a ratio of 1 g of dried leaves for 10 mL of of 80% ethanol under agitation at room temperature for 24 h. The filtrate solutions were evaporated under reduced pressure (Buchi rotavapor R-114, Paris, France) and yield61 ± 2 g for each extract. Crude extracts (1 g) were submitted to a liquid–liquid extraction using 200 mL of a biphasic mixture composed of M1 and M2 (M1: methyl tert-butyl ether: water, 75:25; M2, water: methanol, 75:25), to obtain nine polar and nine apolar fractions. The 18 fractions were dried under reduced pressure and aliquoted for UHPLC–HRMS profiling at 2 mg/mL in a MeOH/Water 80/20 of LC–MS grade (Fisher Scientific, Schwerte, Germany) and for antileishmanial bioassays at 10 mg/mL in DMSO (Sigma-Aldrich, 99.5%, St. Louis, MO, USA).

### 4.3. Purification of Compound ***1***

To purify compound 1, 20 g of crude extract was partitioned between M1:M2 at 1:1 ratio to obtained 11.34 g of apolar fraction. Then, 6.39 g of the apolar fraction was separated using flash chromatography instrument (Spot Ultimate, Armen, France) on a silica column (CHROMABOND ^®^ Flash RS 80 SiOH 40–63 µm) and eluted using a gradient of dichloromethane and MeOH (100% CH_2_Cl_2_ to CH_2_Cl_2_/MeOH 8/2 then 100% MeOH), to obtain six fractions. Fraction 5 (1.33 g) was selected based on its LC–MS profile and chromatographed on a silica column (CHROMABOND ^®^ Flash RS 25 SiOH) with a gradient of chloroform and methanol (100% CHCl_3_ to CHCl_3_/MeOH 8/2), to yield compound **1** (195 mg, 2.02 g/100 g of dried leaves).

The structure of compound **1** was determined by 1D and 2D NMR spectroscopy on a Bruker 500 MHz (Avance 500, Billerica, MA, USA) using dimethylsulfoxide-d6. Chemical shifts (relative to tetramethylsilane (TMS)) are in ppm, and coupling constants in Hz. The compound was identified as “corosolic acid” by comparing with data from the literature [40] (Appendix A).

### 4.4. UHPLC–HRMS Profiling

All extracts (2 mg/mL) were profiled using a UHPLC-DAD-LTQ Orbitrap XL instrument (Ultimate 3000, Thermo Fisher Scientific, Hemel Hempstead, UK). The UV detection from 210 to 400 nm was performed with a diode array detector (DAD) (Hemel Hempstead, UK). Mass detection was performed using an atmospheric pressure chemical ionization (APCI) source in both NI and PI modes at 15,000 resolving power (full width at half maximum at 400 *m*/*z*). The mass scanning range was *m*/*z* 100–1500 Da. The capillary temperature was 300 °C and the spray voltage was fixed at 3.0 kV. Mass measurements were externally calibrated before starting the experiment. Each full MS scan was followed by data-dependent MS/MS on the four most intense peaks using collision-induced dissociation (35% normalized collision energy, isolation width 2 Da, activation Q 0.250). The LC–MS system was run in binary gradient mode using a BEH C18 Acquity column (100 × 2.1 mm i.d., 1.7 µm, 130 Å, Waters, MA, USA) equipped with a guard column. Mobile phase A (MPA) was 0.1% formic acid (FA) in water and mobile phase B (MPB) was 0.1% FA in acetonitrile. Gradient conditions were: 0 min, 95% MPA; 0.5 min 95% MPA; 12 min, 5% MPA; 15 min, 5% MPA, 15.5 min, 95% MPA, and 19 min, 95% MPA. The flow rate was 0.3 mL/min, column temperature 40 °C, and injection volume was 2 µL.

### 4.5. Data Processing

The UHPLC–HRMS raw data were converted to abf files (Reifycs, Japan) and processed with MS-DIAL version 3.90 [42] for mass signal extraction between 100 and 1500 Da from 0.5 to 16.5 min. Respective tolerances for MS1 and MS2 were set to 0.01 and 0.2 Da in centroid mode. The optimized detection threshold was set to 10^5^ (negative) and 2 × 10^5^ (positive) for MS1. The peaks were aligned on a quality control sample (an aliquot of each fraction) reference file with a retention time tolerance of 0.15 min and a mass tolerance of 0.025 Da. Adducts and complexes were identified to exclude them from the final peak list along with features from blanks sample (injection of dilution solvent). Additionally, features with a relative standard deviation above 30% in QC sample were also deleted. The resulting peak list was then exported to comma-separated value (CSV) format prior to multivariate data analysis using SIMCA-P+ (version 15.0.2, Umerics, Umea, Sweden).

### 4.6. Statistical Analysis

CSV files were directly imported into SIMCA-P+ (version 15.0.2, Umerics, Umea, Sweden). For multivariate data analysis, all data were pareto scaled. The OPLS regression analysis was carried out with IC_50_ of antileishmanial activity as the Y input. Coefficient scores were used to rank variables according to their antileishmanial potential.

### 4.7. Identification of Significant Features

Molecular formulae of significant features were calculated with MS-FINDER 3.24 [18]. Various parameters were used in order to reduce the number of potential candidates, such as exclusive selection of the elements C, H, and O; mass tolerance fixed to MS1:0.01 Da and MS2:0.2 Da; and the isotopic ratio tolerance set to 20%. Only natural product databases focused on plants were selected from the DNP (CRC press, v27.2). Compounds from the *Psidium* genus or Myrtaceae family were prioritized. The results were presented as a list of compounds sorted according to the score value of the match. This value encompassed uncertainty on accurate mass, the isotopic pattern score, and the experimental MS/MS fragmentation mirrored to in silico matches. Only chemical identities with a final score above five were retained.

If available, commercial authentic standards were ordered from Wuhan ChemFaces Biochemical Co., ltd. (Wuhan, Hubei, China) to confirm identity. Compound **2** (jacoumaric acid), **5** (guajadial B), and **6** (medicagenic acid) were ordered with a purity ≥ 98% (measured by ^13^C-NMR spectroscopy and ^1^H-NMR spectroscopy).

### 4.8. Antileishmanial Evaluation

The *Leishmania* species used in this study was the *L. infantum* strain MHOM/MA/67/ITMAP-263 (CNR *Leishmania*, Montpellier, France) expressing luciferase activity.

#### 4.8.1. Antileishmanial Activity on Promastigotes

We use luciferase assays to evaluate the effect of the tested compounds on the growth of *L. infantum* promastigotes. Briefly, logarithmic phase promastigotes suspended in RPMI 1640 medium supplemented with 10% fetal calf serum, 2 mM l-glutamine, and antibiotics (100 U/mL penicillin, 100 mg/mL streptomycin, and 50 µg/mL geneticin) were incubated at a density of 10^6^ parasites/mL in sterile 96-well plates with crude extract and fractions (diluted at 50 and 25 µg/mL) in duplicate. Amphotericin B (purchased from Sigma-Aldrich) was used as a positive control. After a 72-h incubation period at 24 °C, we examined each well plate under a microscope to detect possible precipitate formation. To estimate the luciferase activity of promastigotes, 80 µL of each well was transferred into white 96-well plates after mild resuspension. Steady Glow^®^ reagent (Promega, Madison, WI, USA) was added according to the manufacturer’s instructions, and plates were incubated for 2 min at room temperature. The luminescence was measured by a MicroBeta luminescence counter (PerkinElmer, Waltham, MA, USA). For the IC_50_ evaluation, the most active fractions or compounds were measured by eight dosage dilutions (0.39–50 µg/mL). IC_50_ was calculated by non-linear regression analysis processed on dose-response curves, using the GraphPad Prism 6.0. (San Diego, CA, USA). IC_50_ values represent the mean value calculated from three independent experiments.

#### 4.8.2. Antileishmanial Activity on Axenic Amastigotes

Logarithmic phase *L. infantum* promastigotes were centrifuged at 900 g for 10 min. The supernatant was removed carefully and was replaced by the same volume of RPMI 1640 complete medium at pH 5.4 and incubated for 24 h at 24 °C. The acidified promastigotes were incubated for 24 h at 37 °C in a ventilated flask and were transformed into amastigotes. The effect of the tested compounds on the growth of *L. infantum* axenic amastigotes was assessed as follows: *L. infantum* amastigotes were incubated at a density of 2 × 10^6^ parasites/mL in sterile 96-well plates with crude extract and fractions (diluted at 50 and 25 µg/mL) in duplicate. Amphotericin B was used as a positive control. After a 48-h incubation period at 37 °C, each well plate was then examined under a microscope to detect any precipitate formation. Subsequently, to determine the IC_50_, we used the same method described in Section 4.8.1.

### 4.9. Cytotoxicity Evaluation

An MTT (3-(4,5-Dimethylthiazol-2-yl)-2,5-Diphenyltetrazolium Bromide) assay with the J774A.1 cell line (mouse macrophage cell line, Sigma-Aldrich) was performed to evaluate the cytotoxicity of the tested compounds. Briefly, cells (5 × 10^4^ cells/mL) in 100 µL of complete medium (DMEM high glucose supplemented with 10% fetal calf serum (FSC), 2 mM l-glutamine and antibiotics (100 U/mL penicillin and 100 µg/mL streptomycin)) were seeded into each well of 96-well plates and incubated at 37 °C and 5% CO_2_. After incubation for 24 h, 100 µL of medium with various concentrations of jacoumaric acid, carasolic acid, and appropriate controls (doxorubicin and amphotericin B, purchased from Sigma-Aldrich) were added and the plates were incubated for 72 h at 37 °C and 5% CO_2_. Each 96-well plate was then examined under a microscope to detect possible precipitate formation before the medium was aspirated from the well. 100 µL of MTT solution (0.5mg/mL in DMEM) was then added to each well. Cells were incubated for 2 h at 37 °C and 5% CO_2_. Then, the MTT solution was removed and DMSO (100 µL/well) was added to dissolve the resulting formazan crystals. Plates were shaken vigorously (300 rpm) for 5 min. The absorbance was measured at 570 nm with a microplate spectrophotometer (Eon Bio Tek, Winooski, VT, USA). DMSO was used as blank. CC_50_ values were calculated by non-linear regression analysis processed on dose-response curves, using GraphPad Prism 6.0. (San Diego, CA, USA). CC_50_ values represent the mean value calculated from three independent experiments. 

### 4.10. Determination of Intracellular ROS Generation

ROS levels in treated and untreated *L. infantum* amastigotes were monitored using the cell permeable fluorogenic dye H_2_DCFDA [43]. 1 × 10^6^ cells/mL amastigotes were treated with IC_50_ and IC_90_ doses of jacoumaric acid and corosolic acid for 24 h. The cells were then centrifuged, washed with PBS, resuspended in PBS, and incubated for 30 min in the dark with 10 µM H_2_DCFDA at 37 °C. ROS were measured as an increase in fluorescence caused by the conversion of non-fluorescent dye to highly fluorescence H_2_DCFDA (excitation wavelength 490 nm, emission wavelength 525 nm) in a fluorescence microplate reader (SAFAS Xenius XM, Monaco, France).

## 5. Conclusions

We set up a metabolomic workflow to rapidly decipher antileishmanial compounds from *P. guajava* L. leaves extract. Our dereplication approach lead to the identification of corosolic acid and jacoumaric acid as the most active compounds against amastigotes *L. infantum,* along with the ROS scavenging in the antileishmanial mechanism. This is the first report in antileishmanial activity of corosolic acid and jacoumaric acid. Further studies would be required to understand the mechanism of action of these triterpenoids’ compounds on leishmaniasis parasites.

## Figures and Tables

**Figure 1 molecules-24-04536-f001:**
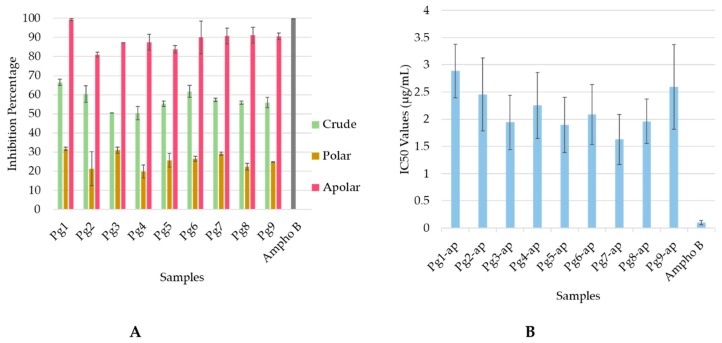
Antileishmanial activity of guava leaf extracts: (**A**) Inhibition percentage at 25 µg/mL of crude extracts and of the corresponding polar and apolar fractions measured on *L. infantum* amastigotes. (**B**) IC_50_ of nine apolar fractions from *P. guajava* (Pg1–9) compared to amphotericin B (Ampho B).

**Figure 2 molecules-24-04536-f002:**
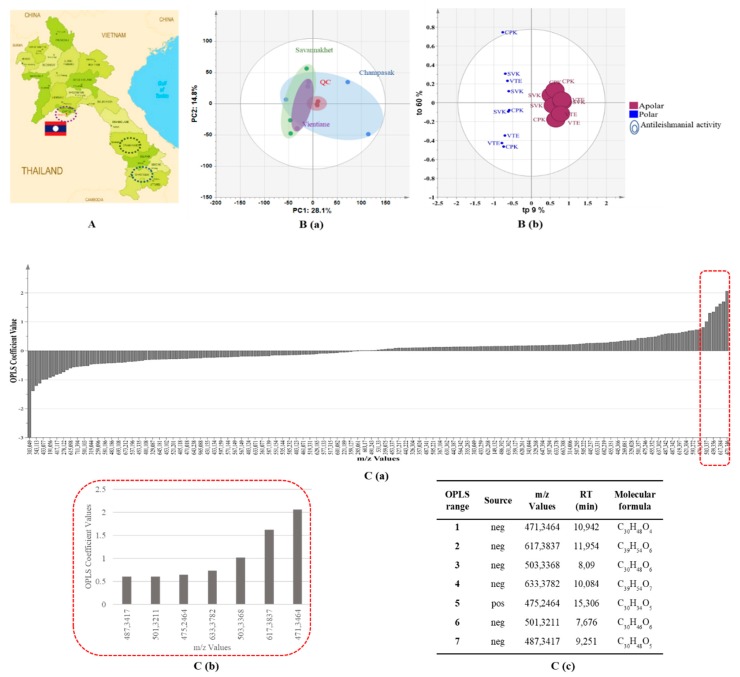
Multivariate data analysis workflow: **A**
*P. guajava* collection areas. **B** (**a**) PCA (principal component analysis) score plot of the APCI-NI&PI (atmospheric pressure chemical ionization – negative and positive ionization) dataset colored according to the collection areas; (**b**) OPLS regression (orthogonal projections to latent structures) score plot of the polar and apolar phase dataset correlated to their antileishmanial activity (High circle radius for low IC_50_). **C** (**a**) Coefficient plot obtained by OPLS regression; (**b**) Emphasis on the first loadings; (**c**) Details of the first seven features according to the putative antileishmanial activity. CPK, Champasak; QC, quality control; SVK, Savannakhet; VTE, Vientiane.

**Figure 3 molecules-24-04536-f003:**
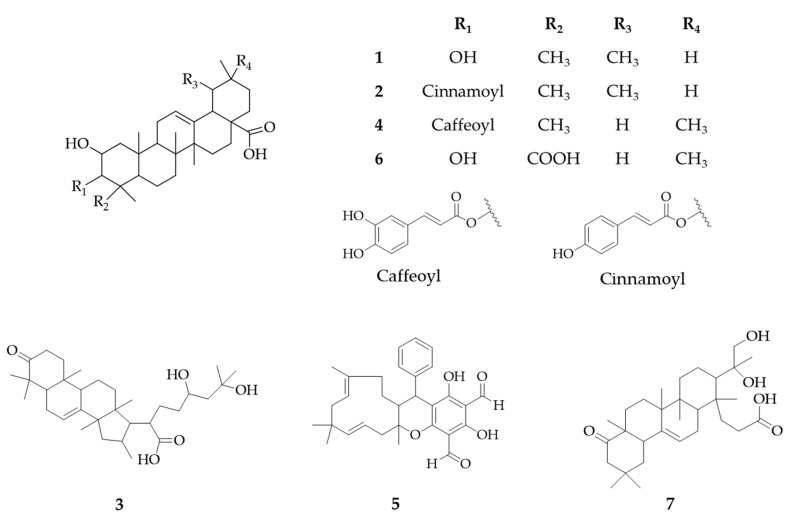
Chemical structures of annotated compounds (See Table 1 for details).

**Figure 4 molecules-24-04536-f004:**
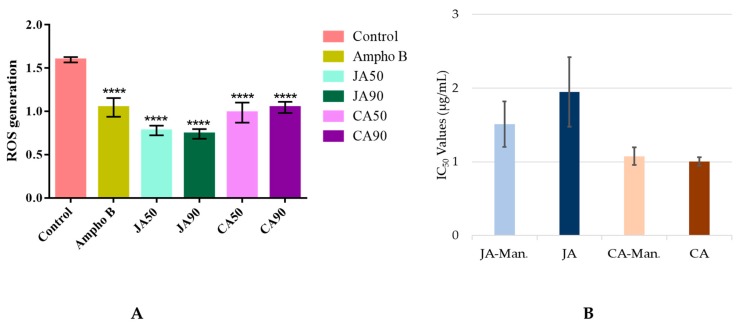
Antioxidant capacity of jacoumaric acid and corosolic acid (JA) and (CA): **A** Reactive oxygen species (ROS) measured using the H_2_DCFDA (2′,7′-dichlorodihydrofluorescein diacetate) assay on axenic amastigotes in the presence of JA and CA at IC_50_ and IC_90_ concentrations compared to amphotericin B (AmphoB) (2 µg/mL). **** indicates significant difference relative to the control group (*p* < 0.0001)). **B** IC_50_ comparison of JA and CA when associated with mannitol 50 µM (JA-Man and CA-Man) (JA: *p* = 0.816 and CA: *p* = 0.403).

**Table 1 molecules-24-04536-t001:** Summary of all the compounds identified or dereplicated in this work.

OPLS Rank *	Source	*m*/*z* Values	RT (min)	Adduct Type	MF	Δ*m*/*z* (mDa)	Main MS/MS Fragments	UV (nm)	Putative Annotation	Chemical Class	Biology Source
**1**	neg	4,713,464	10.942	[M − H]^−^	C_30_H_48_O_4_	1.5836	451.2516 393.3562 289.2787	ND	Corosolic acid ^a,c^ (2α-hydroxyursolic acid)	Ursane triterpenoids	*Lagerstroemia speciosa* [19]
**2**	neg	6,173,837	11.954	[M − H]^−^	C_39_H_54_O_6_	1.063	597.0866 537.0319 497.4617 453.3008	310	Jacoumaric acid ^a,b^	Ursane triterpenoids	*Jacaranda caucana* [20]
**3**	neg	5,033,368	8.090	[M − H]^−^	C_30_H_48_O_6_	1.0128	485.3478 437.4027 389.4317 183.3482	ND	16,24,25-Trihydroxy-3-oxoeuph-7-en-21-oic acid ^a^	Dammarane triterpenoids	*Psidium guajava*
**4**	neg	6,333,782	10.084	[M − H]^−^	C_39_H_54_O_7_	1.4776	563.7110 513.4430 469.5226 392.2423	300	2,3-Dihydroxy-12-oleanen-28-oic acid; 3-*O*-(3,4-Dihydroxy-E-cinnamoyl) ^a^	Oleanane triterpenoids	*Psidium guajava*
**5**	pos	4,752,464	15.306	[M + H]^+^	C_30_H_34_O_5_	1.5006	323.2012 293.3159 271.0567 205.2643	278	Guajadial B ^a,b^	Meroterpenoids	*Psidium guajava* [21]
**6**	neg	5,013,211	7.676	[M − H]^−^	C_30_H_46_O_6_	1.0627	483.6488 389.3652 321.4315 189.4521	ND	Medicagenic acid ^a,b^	Ursane triterpenoids	*Medicago sativa* [22]
**7**	neg	4,873,417	9.251	[M − H]^−^	C_30_H_48_O_5_	1.1982	469.5041 424.4036 268.196	ND	4,23-Dihydroxy-22-oxo-3,4-seco-12-oleanen-3-oic acid ^a^	Oleanane triterpenoids	*Streptomyces* sp.

* Ranking based on PLS regression coefficients, served as compound number. ^a^ Determined by in silico MS/MS fragmentation with MS-FINDER. ^b^ Confirmed by commercial authentic standard compounds. ^c^ Confirmed by NMR spectroscopy. ND, not detected; NMR, nuclear magnetic resonance; MF, molecular formula; MS/MS, tandem mass spectrometry; OPLS, orthogonal partial least squares/projections to latent structures; UV, ultraviolet.

**Table 2 molecules-24-04536-t002:** Confirmation of the antileishmanial potential of putative annotations.

Compounds	Promastigotes	Axenic Amastigotes	J774A1	SI	
	IC_50_ (µg/mL)	IC_50_ (µg/mL)	CC_50_ (µg/mL)	Promastigotes	Amastigotes
**Pooled apolar fraction**	>50	1.96 ± 0.47	51.19 ± 9.21	ND	26.12
**Corosolic acid (1)**	18.43 ± 1.20	1.01 ± 0.06	5.77 ± 0.50	0.31	5.71
**Jacoumaric acid (2)**	>50	1.32 ± 0.59	12.88 ± 2.50	ND	9.76
**Guajadial B (5)**	>50	>50	NT	ND	ND
**Medicagenic acid (6)**	>50	>50	NT	ND	ND
**Amphotericin B**	0.09 ± 0.01	0.09 ± 0.04	5.79 ± 0.65	64.33	64.33
**Doxorubicin**	NT	NT	0.04 ± 0.004	ND	ND

NT, None Tested; ND, not Detected; SI, Selectivity Index.

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
