# Peer review of "Antileishmanial Compounds Isolated from Psidium Guajava L. Using a Metabolomic Approach"

_molecules, 2019, doi:10.3390/molecules24244536_

Round 1

Reviewer 1 Report

Reviewer’s criticism to MOLECULES-667001 manuscript

In this study, authors investigated the antileishmanial activity of natural products of Psidium guajava. Several crude extracts from P. guajava leaves was collected from different region and the common components of P. guajava were determined by LC-HRMS and highlighted their antileishmanial activity by multivariate correlation analysis. The whole manuscript is well described and high quality. Use of the metabolic workflow to identification of potent candidates is appropriate and especially valuable. I have only some comments and suggestions.

Therefore, I can recommend this paper for publication in Molecules.

Detailed comments and questions:

1; Line 30

Delete one of “steps” word.

2; Figure 1

This Figure is completely missing.

3, Line 118, Figure S2

Authors should added some fragmentation results of the standards and compare the fragmentation profiles of candidates to these not only the in silico fragmentation profiles.

“If available, commercial standards were ordered from Wuhan ChemFaces Biochemical Co., ltd. 329 (Wuhan, Hubei, China) to confirm identity. Compound 2 (jacoumaric acid), 5 (guajadial B), and 6 330 (medicagenic acid) were ordered…

and

b Confirmed by commercial standard compounds”

4; section 4.3

Line 231 6.39 g

Line 234 Fraction 5 (1.33 g)

Line 236 compound 1 (195 mg)

Why important data are the weight of different compound, fraction? How could you measured it? How many separated fractions were observed and how could you select the relevant? Authors should introduce and specify these.

Based on these critical issues, reviewer can recommend the manuscript for publication in the Molecules, but minor revision is needed.

Reviewer 2 Report

This manuscript describes the development of a method for the analysis and identification of selected secondary plant metabolites of the leaves of the tropical tree Psidium guajava Linn., based on an ethanol extraction of the dried leaves with subsequent liquid-liquid extraction, fractionation on silica followed by analysis with ultra high performance liquid chromatography high resolution mass spectrometry and multivariate correlation analysis. Compounds were identified with tandem mass spectrometry and compared with in silico fragmentation data with identifications cross-validated with authentic reference standards. The bioactivity of individual fractions against the protozoan parasite Leishmania was evaluated for a species that had luciferase activity, allowing the quantitation of live cells with luminescence detection. In addition, the toxicity of tested compounds was evaluated with an MTT cytotoxicity test. The described approach has allowed the identification of several natural product compounds from the class of triterpenoids that showed antileishmanial activity. The manuscript is reasonably well written, with an appropriate experimental design. The results are clearly presented with the conclusions supported by the results. However, before a publication in Molecules can be considered, it is recommended to revise the Introduction and the Material and Method sections to enhance the readability of the manuscript and to correct typographical errors.

Comment 1: The legends of Figures 1 and 2 do not match the Figures and presumably need to be swapped. 

Comment 2: It is suggested to provide a little more background on the epidemiology of leishmaniasis in the introduction on Page 1, Lines 41 after the sentence ending with …. phlebotomine sandfly.”, for example: “The epidemiology of leishmaniasis depends on a variety of factors including the characteristics of the parasite and sandfly species, the local ecological characteristics of the transmission sites, exposure of the human population to the parasite, and human behavior.”.

Comment 3: It recommended to provide the pore size of the reversed-phase column in the Material and Method section.

Comment 4: It recommended to provide a conclusion section for the manuscript.

Comment 5: The use of the nomenclature and English language could be improved e.g.:

Please italicize “m/z” throughout the manuscript.

Please italicize all species names throughout the manuscript.

Please italicize “in vitro” throughout the manuscript.

Page 1, Line 17: Please replace “treatments” with “medical treatments”.

Page 1, Line 19: Please replace “treatments” with “cure”.

Page 2, Line 86: Please replace “locations” with “locations in the”.

Page 1, Line 38: Please replace “low-income” with “low socioeconomic”.

Page 1, Line 41: Please replace “mammalian animals” with “mammalians”.

Page 1, Line 43: Please replace “treatments” with “medical treatments”.

Page 2, Line 50: Please replace “inexpensive molecules” with “cost-effectively produced bioactive molecules”.

Page 2, Line 55: Please replace “activity of a promising plant species” with “activity of natural compounds from a promising plant species”.

Page 2, Line 68: Please replace “Our groups” with “Our group”.

Page 1, Line 43: Please replace “commercial standards” with “commercial authentic standards”.

Page 5, Line 145, Page 8, Line 237 and Page 10, Line 331: Please replace “H-NMR” with “H-NMR spectroscopy”.

Page 8, Line 209: Please replace “treterpenoid” with “triterpenoid”.

Page 8, Line 220: Please replace “voucher specimen” with “sample specimen”.

Page 8, Line 240: Please replace “litterature” with “literature”.

Page 9, Line 247: Please replace “measurement was” with “measurements were”.

πPage 15, Line 480: Please replace “……” with relevant compound names.

Reviewer 3 Report

This is a review on the manuscript entitled "Antileishmanial compounds isolated from Psidium guajava L. using a metabolomic approach". In this paper authors obtain extracts from guajava and correlate the results of toxicological test on leishmania with fingerprints obtained by LC-MS in order to identify the most active compounds as well as their activity mechanisms. The methodology is not new as authors have used it other times, but the application to guajava and leishmaniosis assures the novelty in the paper and makes it suitable for Molecules, specially in the natural products chemistry section. Overall the paper has a high quality, but the shape of the manuscript requires further polishing. I would suggest that this paper is accepted after correcting some structure problems.

The structure looks to have suffered some copy-paste issues. The experimental part is at the end of the paper. Is should be properly placed. Figures are missing. Principal sections lack also some order as e.g. the experimental section present the toxicological tests before and after LCMS, not grouped… Please check the structure.

I understand that the power of the proposed method relies on the variability of concentrations of single components in the extracts of different origins, but what is the objective of fractionating each extract into 2 polar/apolar fractions?, and why to do it manually? Is there any statistical advantage? Are compounds of interest equilibrating between the two phases, or only in one? Some other publications fractionate the compounds with preparative HPLC into several tens of fraction in order to isolate the target compounds, but I understand that in this case, the power of the method would come only from the different proportions according to their origin. So please, answer to this question: Could the same method be applied without fractionation, that is, simply using extracts from different origins? If so, what is the advantage of the proposed method, and could it be done fractionating into e.g. 30 different fractions by preparative HPLC + evaporation and reconstitution?

Line 219. How were the leaves dried? I guess it was air-dried, protected from sunlight. Please explicit.

It was difficult to follow what happens with the initial mass of leaves. Could you please explicit in the different sections how much mass of leaves or extracts is used? It starts with 250 g of leaves, and the only further quantification was that 2 uL were injected in the LCMS. Where is the sample going?

The extraction section will be difficult to reproduce. Could you explicit the composition of the extractants? 1:10 of 80%  EtOH means 10 g of leaves, 72 g of EtOH and 18 of H2O? The same with the polar-apolar fractionation. Both M1 and M2 contain water. This means that it was a tricomponet mixture with 2 phases? Please explicit.

Line 233. The gradient is confusing 100%DCM > 80%DCM 100%MeOH ?? The second part does not sum up to 100%. How was the gradient made? 6 different mixtures were prepared and manually perfused? Purification of compound 1 should not be here, it has not even been mentioned.

Line 310. Alignent is mentioned, but not if other filtration strategies were used. E.g. only features present in all samples and not in the blanks were selected.

I guess figure 2 should be S2. I guess table 1 should be S1.

Line 83. 8 fractions? There were 9 initially.

Line 85. What are the 25 and 50 ug/mL referring to? Extract? Fraction? Original leaves? Suspect compound? What are the units of the IC50 values?

Figure 1 is missing.

Line 109. What is the structure of the Y data? What information does it consist of?

If the stardards of corosolic acid are commercial, why do authors isolate it from the extracts, and perform RMN? Is only to prove the feasibility of extraction and purification at large scale?

If corosolic and jacoumaric acid are the most active compounds, are authors scheduling to repeat this work with Banabá and Jacaranda?

In figure 3, R4 and R5 are equal for all the structures. This could be simplified. Formulas could be included in Table 1.

In table 2, where does the Amphotericin B data come from? It seems to be 10 times more powerful than the studied compounds.

Since authors have acces to the corosolic and jacoumaric acid, it would be nice if they could provide an approximation of the concentrations of these acids in the extracts or raw leaves.

line 68 groups have or group has.

References are updated (65% less than 5 years) and are of broad origin.
